

# Statistical reconstruction of daily temperature and sea-level pressure in Europe for the severe winter 1788/9

Duncan Pappert[1,2], Mariano Barriendos[3], Yuri Brugnara[1,2], Noemi Imfeld[1,2], Sylvie Jourdain[4],

Rajmund Przybylak[5,6], Christian Rohr[2,7], and Stefan Brönnimann[1,2]

[1]Institute of Geography, University of Bern, Bern, Switzerland
[2]Oeschger Centre for Climate Research, University of Bern, Bern, Switzerland
[3]Department of History and Archaeology, University of Barcelona, Barcelona, Spain
[4]Direction de la Climatologie et des Service Climatiques, Météo-France, Toulouse, France
[5]Faculty of Earth Sciences and Spatial Management, Nicolaus Copernicus University, Torun, Poland´
[6]Centre for Climate Change Research, Nicolaus Copernicus University, Torun, Poland´
[7]Institute of History, University of Bern, Bern, Switzerland

Correspondence: Stefan Brönnimann (stefan.broennimann@giub.unibe.ch)

**Abstract.** The winter 1788/9 was one of the coldest winters Europe had witnessed in the past 300 years. Fortunately for
historical climatologists, this extreme event occurred at a time when many stations across Europe, both private and as part
of coordinated networks, were making quantitative observations of the weather. This means that several dozens of early
instrumental series are available to carry out an in-depth study of this severe cold spell. While there have been attempts to
present daily spatial information for this winter, there is more to be done to understand the weather variability and day-to-
day processes that characterised this weather extreme. In this study, we seek to reconstruct daily spatial high-resolution
temperature and sea level pressure fields of the winter 1788/9 in Europe, from November through February. The
reconstruction is performed with an analogue resampling method (ARM) that uses both historical instrumental data and a
weather type classification. Analogue reconstructions are then post-processed through an ensemble Kalman fitting (EnKF)
technique. Validation experiments show a good skill for both reconstructed variables, which manage to capture the dynamics
of the extreme in relation to the large-scale circulation. These results are promising for more such studies to be undertaken,
focusing on different extreme events and other regions in Europe and perhaps even further back in time. The dataset
presented in this study may be of sufficient quality to allow historians to better assess the environmental and social impacts
of the harsh weather.

## 1  Introduction

The climate system displays variability on interannual to multidecadal time scales, which in turn affects the frequency and
intensity of extreme weather in ways which are not yet fully understood. Such extremes have been observed to occur in the
past and will reoccur in the future with high impact on environmental, social and economic conditions. Early instrumental
meteorological observations hold enormous value in climate research to gain a better understanding of past climate
variability and the mechanisms that characterise extreme weather events.



Recent decades have seen a number of different projects put much time and effort into the rescue of some of these data,

including making inventories, digitising, converting, correcting and finally adding these instrumental series into global data repositories (Camuffo and Jones, 2002; Vinther et al., 2006; Auer et al., 2007; Cornes, 2010; Csernus-Molnár et al., 2014; Cram et al., 2015; Brönnimann et al., 2019; Pappert et al., 2021). Data rescue projects and activities are facilitated by collaboration with initiatives such as the Atmospheric Circulation Reconstructions over the Earth (ACRE) project (Allan et al., 2016). These data rescue efforts feed into – or are often triggered by – historical reanalysis projects (Slivinski et al., 2019).

Recovering these data is challenging and there are many old weather records still waiting to be digitised and consolidated. Early instrumental data for Europe are scattered around hundreds of libraries and archives across the continent in the form of countless documents and diaries. Their collection and digitisation require significant time, financial and human resources, which goes some way towards explaining why many of such series have yet to be analysed. In some cases, the absence of metadata, for instance specific information regarding the position of instruments and method of observation, might impact

the quality of the series; this requires assumptions to be made that may increase the uncertainty of the original observations (Brugnara et al., 2015). This study will show how many of the instrumental series could have promising applications in scientific research despite such problems. These series can be combined with the possibilities offered by today's dynamical and stochastic models, including various data assimilation and statistical downscaling techniques, to offer more complete spatial information on past weather. The daily time scale of such weather reconstructions will help better understand the

underlying atmospheric processes associated with extreme events. Moreover, current historical reanalyses go back to the 19th century, but not further. In this paper, we demonstrate another approach for continental-scale weather reconstruction using as a case study a late 18th-century extreme weather event.

The extreme event at the centre of this study is the severely cold winter of 1788/9. It ranks amongst the harshest winters in Europe in the past 300 years with perhaps the coldest December for many regions, especially Central Europe (Brázdil et

al., 2003). Despite the severity of this extreme, the winter 1788/9 has received only modest attention in climate research, though important work has been done towards characterising this event (see Kington, 1980; Gisler, 1985; Barriendos et al., 2000; Brázdil et al., 2003; Pfister et al., 2019). Perhaps the most detailed study of this winter was made by Barriendos et al. (2000), who reflected on the extensive damage it inflicted on human communities and analysed the event at a daily resolution using instrumental data for the Iberian Peninsula. Indeed, the 1780s saw numerous stations in Europe making

quantitative observations of the weather. In his book *The Weather of the 1780s over Europe*, Kington (1988) used the available station data from this decade to create hand-drawn daily synoptic weather maps for this period. His experiment showed that prevailing weather situations over Europe at the time could be reconstructed and analysed; though, unfortunately, the maps for the period after 1785 were not published. Since Kington, scarcely any statistical reconstructions of daily weather fields for this period have been attempted, mainly because of the challenges presented by the gathering of

these data from numerous different sources and their relative higher uncertainty compared to postindustrial measurements. Following thus in Kington's steps, this study seeks to make use of the remarkable volume of instrumental observations that



exist for this period to reconstruct daily weather fields over Europe of the winter 1788/9 in an attempt to better understand the dynamics that characterised such an outstanding weather extreme.

For this article, the method used for the reconstruction is as important as the choice of the extreme event itself. Historical
station data capture local conditions and are mostly not representative of the regional weather. To overcome this hurdle, we use a statistical approach that has recently been used in weather reconstructions based on historical data: the analogue resampling method (ARM) (Flückiger et al., 2017; Rössler and Brönnimann, 2018) – a variation of the more classical analogue method used in forecasting (Lorenz, 1969; Kruizinga and Murphy, 1983; Zorita and Storch, 1999; Horton et al., 2012). The ARM is based on the assumption that similar patterns of atmospheric states repeat themselves throughout time producing
similar local effects. Two states "which are observed to resemble one another are called analogues" (Lorenz, 1969, p. 636). This statistical relationship between local weather and synoptic patterns allows the analogue approach to use one to predict the other. Thus the ARM makes predictions for historical weather patterns by searching records from the recent present for analogous weather patterns to those that occurred in the past, thereby resampling known weather fields for which there is detailed spatial information. In this study, temperature analogues are sampled from daily 0.1° resolved fields from the E-OBS
dataset (Cornes et al., 2018) and for sea level pressure (SLP) from 0.25° fields from ERA5 (Hersbach et al., 2020). These fields are further improved upon in a post-processing procedure that uses a data assimilation technique known as the ensemble Kalman filter (EnKF).

This study builds on the work of Pfister et al. (2020), who use the ARM to generate daily gridded meteorological fields for Switzerland. In their paper, they demonstrate the good skill of the ARM in reconstructing daily temperature and precipitation
fields over Switzerland back to 1864. They conclude with the suggestion to extend the reconstruction further back in time to the preindustrial period. Additionally, they address the possible suitability of the method for Central Europe and beyond. We therefore also perform a validation to assess the ability of the approach to reproduce synoptic conditions over Europe for the winter 1788/9. Results will show the skill of the reconstructions, which will in turn be explored to assess if they are in line with what is already known about the weather in Europe during this season and if they allow us an even more detailed insight
into the progression of the cold spell.

A successful reconstruction of the winter 1788/9 would allow for comparisons with other more recent cold temperature extremes. In fact, this study could be seen as a continuation or expansion of the literature on extremely cold winters in Europe. Important work on this topic has been done to study the frequency, timing, intensity, and spatial extent of extremely cold winters (ECWs) and extremely cold winter months (ECMs) for the period 1951-2010 (Twardosz and Kossowska-Cezak,
2016; Twardosz et al., 2016). Our study could open a new chapter for the study of pre-industrial extreme cold winters using high spatial and temporal resolution gridded data; these could then be compared to severe winters in the modern era to acquire a better understanding of extreme weather variability.

The next section introduces the datasets and the methodology used for the reconstruction, validation and analysis. In Sect. 3, we present and discuss the validation of the ARM-reconstructed and post-processed fields of temperature and sea level
pressure. The last part of this section examines the current state of knowledge on the winter 1788/9 and explores some of



the ways in which the reconstructed maps can improve our understanding of the extreme event. The final section makes concluding remarks about the study.

## 2    Data and Methods

### 2.1    Data

To predict a certain aspect of the atmosphere, here for instance, temperature or SLP fields for a given day of interest in the past (predictand), the analogue approach searches for the day in the reference period with the most similar predictor values (best analogue) and borrows the spatial atmospheric pattern from this (or multiple) best analogue day(s) as a prediction (Zorita et al., 1995). In the case of the present study, differences between states of the atmosphere are measured in terms

of the differences between two sets of series, one historical and the other a reference for which spatial data exists. In line with this explanation, the ARM thus requires three archives: historical station data, reference station data, and gridded datasets from which the spatial information for the analogues is resampled. The following subsections summarise these datasets as used for the reconstruction of the winter 1788/9.

### 2.1.1    Historical meteorological data

Table 1 lists the stations whose observations were used for the statistical reconstruction. These comprise 52 temperature and 41 SLP series from a total of 53 stations, of which 8 have data for only either 1788 or 1789. Independent measurements from 2 additional stations were used for validation. Figure 1 shows how these stations are distributed across Europe: most are concentrated in Central and Western Europe, with only very sparse coverage in the North, South, and East. While these do not represent all stations making measurements in Europe during these years (see Brönnimann et al., 2019), they

constitute a large portion of known observation series. Unfortunately, there are virtually no known early instrumental observations for the South-East during this period. Gathering good quality instrumental series is an onerous task and requires delving into dozens of regional archives, collecting any relevant metadata (which is often incomplete or absent), and digitising the observations, which then undergo various quality control tests and corrections. Of the series used in this study, some had already been analysed and corrected (Camuffo and Jones, 2002; Cornes, 2010; Parker et al., 2014; Pfister et al., 2019;

Pappert et al., 2021), some had been investigated for their metadata and digitised (Barriendos et al., 2000; Mazón et al., 2011; Csernus-Molnár et al., 2014; Dawson et al., 2021), while others had yet to be digitised and explored. As such, the series differ in degrees of quality, not only for reasons related to how they were observed in the past, but also for the extent of the attention they have received by researchers today.



**Table 1:** List of stations used for the reconstruction, described by the modern country they belong to, their station code, approximate height above sea level, latitude, longitude, the name of the observer(s) if known, and the authority (private or network). Stations marked by (*) represent those series that are used in the independent validation. SMP, SRM, and RAMB stand respectively for *Societas Meteorologica Palatina, Société Royale de Médécine, and Royal Academy of Medicine of Barcelona*. The last column lists any remarks pertaining to a series: *ta* means that only temperature is used, *slp* for just sea level pressure; 1788 or 1789 means that only data for these

years is available; a blank implies that both variables were used in the reconstruction across both years.

| Station | Country | Abbr. | m.a.s.l. | Lat | Lon | Observer | Authority | Remarks |
|---|---|---|---|---|---|---|---|---|
| Barcelona | Spain | BAR | 35 | 41.39 | 2.16 | Salva | RAMB | |
| Basel | Switzerland | BAS | 275 | 47.56 | 7.57 | d'Annone | Private | |
| Berlin | Germany | BRL | 45 | 52.52 | 13.4 | Beguelin | SMP | 1788 |
| Bern | Switzerland | BER | 534 | 46.95 | 7.45 | Lombach | Private | |
| Bologna | Italy | BOL | 64 | 44.49 | 11.34 | Cospi | Private | ta |
| Bridestowe | England | BSW | 167 | 50.685 | -4.1 | Heberdem | Private | ta |
| Budapest | Hungary | BUD | 168 | 47.5 | 19.04 | Weiss | SMP | |
| Cadiz * | Spain | CAD | 5 | 36.54 | -6.28 | multiple | Private | gaps |
| Central Belgium T. | Belgium | CBT | - | 50.65 | 4.85 | multiple | various | ta |
| Central England T. | England | CET | - | 52.69 | -1.4 | multiple | various | ta |
| Clermont-Ferrand | France | CLF | 394 | 45.78 | 3.1 | Delarbre | SRM | |
| Copenhagen | Denmark | COP | 40 | 55.68 | 12.57 | Bugge | SMP | 1788 |
| Edinburgh | Scotland | EDI | 40 | 55.85 | -3.175 | McFarland | Private | |
| Erfurt | Germany | ERF | 202 | 50.98 | 11.03 | Planer | SMP | 1788 |
| Geneva | Switzerland | GEN | 380 | 46.2 | 6.14 | Senebier | SMP | |
| Gotthard | Switzerland | GOT | 2093 | 46.55 | 8.57 | Mediolanensi | SMP | |
| Gordon Castle | Scotland | GDC | 40 | 57.62 | -3.09 | Hoy | Private | ta |
| Hohenpeissenberg | Germany | HOH | 977 | 47.8 | 11.01 | Schwaiger | SMP | |
| Innsbruck | Austria | INN | 584 | 47.26 | 11.4 | Zallinger | Private | ta |
| Kezmarok | Slovakia | KEZ | 626 | 49.135 | 20.43 | Genersich | Private | 1789 |
| La Rochelle | France | ROC | 19 | 46.16 | -1.15 | Seignette | SMP | |
| Lille | France | LIL | 25 | 50.63 | 3.06 | Saladin | SRM | |
| Liverpool | England | LIV | 15 | 53.4 | -2.99 | Woodworth | Private | |
| London | England | LDN | 24 | 51.5 | -0.12 | multiple | Private | |
| Lüneburg | Germany | LNB | 25 | 53.26 | 10.41 | Ebeling | Private | 1788 |
| Madrid | Spain | MAD | 680 | 40.42 | -3.7 | Salanova | Private | slp |
| Mannheim | Germany | MAN | 112 | 49.49 | 8.47 | Hemmer | SMP | |
| Marseille | France | MAR | 44 | 43.3 | 5.36 | de Silvabelle | SMP | |
| Middelburg | Belgium | MID | 15 | 51.499 | 3.61 | Van de Perre | SMP | 1788 |
| Milan | Italy | MIL | 132 | 45.46 | 9.19 | anonymous | Private | |
| Montauban | France | MNT | 112 | 44.02 | 1.35 | Moulet | SRM | |
| Moscow | Russia | MOS | 130 | 55.76 | 37.62 | Stritter | SMP | ta |



| Nancy | France | NAN | 215 | 48.69 | 6.184 | Poma Toaldo, | SRM | |
| Padua | Italy | PAD | 18 | 45.41 | 11.88 | Chiminello | SMP | |
| Paris | France | PAR | 74 | 48.86 | 2.35 | anonymous | Private | |
| Prague | Czechia | PRK | 200 | 50.07 | 14.42 | Strnadt | Private | ta |
| Regensburg | Germany | REG | 346 | 49.01 | 12.1 | Heinrich | SMP | |
| Rieux-Volvestre | France | RXV | 220 | 43.25 | 1.2 | Darbas | SRM | |
| Rome | Italy | ROM | 56 | 41.9 | 12.5 | Calandrelli | SMP | |
| Rostock | Germany | RST | 35 | 54.09 | 12.1 | Schadeloock | Private | |
| Rouen | France | ROU | 15 | 49.44 | 1.1 | Lepecq | SRM | |
| Saint-Brieuc | France | STB | 100 | 48.51 | -2.79 | Bagot | SRM | |
| Saint-Paul-Trois-Ch. | France | SPT | 85 | 44.35 | 4.77 | Caudeiron | SRM | gaps |
| St Petersburg | Russia | PET | 15 | 59.93 | 30.36 | Euler | SMP | |
| Stockholm | Sweden | STO | 44 | 59.21 | 18.03 | Nicander | SMP | |
| Stroud | England | STR | 63 | 51.75 | -2.22 | Hughes | Private | ta, 1788 |
| Trondheim | Norway | TRH | 13 | 63.45 | 10.42 | Fester | Private | |
| Turku | Finland | TRK | 10 | 60.45 | 22.28 | Planman | Private | |
| Uppsala | Sweden | UPS | 15 | 59.86 | 17.64 | multiple | Private | |
| Vienna | Austria | VIE | 208 | 48.21 | 16.36 | Kletten | Private | |
| Vilnius | Lithuania | VIL | 118 | 54.69 | 25.28 | Poczobutt-Odlanicki | Private | ta |
| Warsaw | Poland | WAR | 106 | 52.24 | 21.02 | Bończa-Bystrzycki | Private | ta |
| Zagan | Poland | ZAG | 116 | 51.37 | 15.19 | Preuss | SMP | |
| Zurich * | Switzerland | ZUR | 418 | 47.37 | 8.55 | Muralt | Private | |
| Zwanenburg | Netherlands | ZWA | 10 | 52.38 | 4.8 | multiple | Private | |

Stations whose observations had not undergone any prior processing, were converted into modern units and underwent quality control tests provided by the C3S QC software *dataresqc* (see Brugnara et al., 2020; Pappert et al., 2021). Temperature daily means were calculated from subdaily values by taking into account known observations times and adjusted according to the ERA5-Land diurnal cycles for 1981-2010. If observation times were unknown, means were calculated from thrice daily morning-afternoon-evening measurements according to the formula $(t_1 + t_2 + t_3 * 2)/4$, and from twice daily morning-afternoon observations according to $(t_1 + t_2)/2$. Pressure daily means were calculated as an arithmetic average and subsequently reduced to mean sea level. Available metadata pertaining to each series is written in the Meta columns of the Standard Exchange Format (SEF, Brunet et al., 2020) files provided in the electronic supplement.



### 2.1.2    Reference meteorological data

Present-day measurement series covering the period 1950-2020 (predictors) are needed to predict the historical spatial fields. For temperature, data were taken from a number of different sources, mostly from the European Climate Assessment Dataset project (ECA&D), as well as from national weather services such as MeteoSwiss, Météo France, the German
Meteorological Service (DWD) and the Met Office. Given the general difficulty in obtaining high quality, consistent SLP measurement series for the period 1950-2020, data for this variable were taken from the grid cell corresponding to the station location in the E-OBS observational dataset. If a temperature series was a blend of different locations the Craddock test was applied (Craddock, 1979; Brunetti et al., 2006; Venema et al., 2012; Fessehaye et al., 2019) and it was homogenised using the same method as Pappert et al. (2021). Each blended series (candidate) was tested against the corresponding grid
cell from E-OBS (reference) based on their monthly cumulative differences. Constant corrections were applied based on differences between the candidate and reference series before and after the breakpoint(s). As in the case of the historical series, the SEF files for the reference series contain details for each day regarding the source, as well as the extent of the correction if homogenised.

### 2.1.3    Gridded meteorological data

The spatial information for the temperature reconstruction is re-sampled from E-OBS, a daily gridded land-only observational dataset over Europe (Cornes et al., 2018). We use the ensemble mean of Version 23.1e release in March 2021 on a 0.1° regular grid and calculated from 100 ensemble members back to 1950. SLP fields, on the other hand, are drawn from the ERA5 dataset, the fifth generation ECMWF reanalysis for the global climate and weather (Hersbach et al., 2020). We use the recent ERA5 version that includes the preliminary back extension to 1950, gridded on a 0.25° regular grid. The domain for
the gridded reconstruction for both variables extends over Europe from -10° to 40° E and from 36° to 65° N.

### 2.2    Pre-processing

Before applying the ARM, station and gridded data are pre-processed in order to make more accurate comparisons between past and present climate variability, thereby facilitating the search for analogues.

In the 160 or so years that separate the historical and reference data, there have been changes in the station's local
environment or site relocations, instruments and observing practices have also changed. Thus the first pre-processing step is to homogenise the historical data to better reflect the measurement conditions of the series in the reference period and reduce discrepancies between the two sets of measurements. This was done by extracting mean temperature and SLP from the EKF400v2 reanalysis (Franke et al., 2017; Valler et al., 2021) for each grid cell corresponding to the location of each historical weather station. The historical series are then adjusted with a monthly correction so they deviate from the mean
reanalysis curve to the same extent as the reference data, without affecting their daily variability. The assumption is that the EKF400v2 reconstruction manages to capture the monthly variability and the long-term past climatic development of temperature and SLP.



Further, the measurement series contain local influences that are not resolved by the spatial data of the gridded dataset. This was accounted for by subtracting from the station data the monthly bias between the measured values and grid cell values of the corresponding locations from the E-OBS and ERA5 datasets over the period 1950-2020.

Reference station and gridded temperature data are then detrended to remove the climate change signal within the period 1950-2020. For this, a simple linear regression is fitted to ERA5 temperature zonal means over land; this trend is thought to best represent large-scale climate change that is unrelated to changes in the frequency of, e.g., weather types. The fitted values are centred around the mid-period 1985 and subtracted from the reference and gridded data. Additionally, we needed to account for the intercentennial climate change trend: temperature data in our reference period (20th century) are warmer on average relative to the pre-industrial benchmark (18th century). For the daily variability between the two periods to be compared, we therefore needed to deduct this climate change signal from the reference data (both station and gridded), thereby giving it a pre-industrial temperature baseline. This signal was calculated by subtracting the EKF400v2 zonally averaged temperature 1751-1800 over land – assumed to represent the climatology for 1788/9 – from the zonally averaged temperature 1950-2003 over land; again, this trend is thought to represent the change in the large-scale baseline climate that is considered unrelated to synoptic-scale conditions. The resulting difference or 'offset' was then subtracted from the reference temperature data for each latitude point. By doing so, physical consistency is not strictly maintained. Nevertheless, we justify this methodological choice from a theoretical standpoint: the subtraction of the offset was performed for the sake of extrapolation. Without performing this step, the analogue pool would be too warm and have insufficient suitable matches for the colder historical data; the gridded fields, also from the period 1950-2020, would also not reflect the climatology of late 18th-century Europe. In other words, individual days in 1788/9 will find fewer analogues if the intercentennial climate change signal is not taken into account; this could lead to problems for extreme days. A systematic error is not expected.

The ARM procedure requires expressing the weather data as anomalies relative to a mean climatology, i.e. as 'irregular' variation components. For this, the seasonal component is removed from the temperature data. The mean seasonality curve is estimated for each observation series by fitting the first two harmonics of the annual temperature cycle using linear regression according to the following equation:

$$S = c_0 + c_1 \sin\left(\frac{2\pi doy}{n_{doy}}\right) + c_2 \cos\left(\frac{2\pi doy}{n_{doy}}\right) + c_3 \sin\left(\frac{4\pi doy}{n_{doy}}\right) + c_4 \cos\left(\frac{4\pi doy}{n_{doy}}\right) \tag{1}$$

where $doy$ is the day of the year, $n_{doy}$ the number of days in the year and $c_0$, $c_1$, $c_2$, $c_3$ and $c_4$ are the parameters to be estimated. For the spatial fields, the seasonality is removed by subtracting the multi-year daily mean for 1950-2020. Now decomposed into a smoothed mean climatology and the respective anomalies. Finally, because temperature and SLP data have different scales, both variables were standardised to have unit variance. The already deseasonalised temperature data was divided by its standard deviation, whereas the SLP data had its mean subtracted and divided by the standard deviation. The same calculations were applied to the 4-month period in 1788/9 using the parameters of the reference period.





### 2.3    Analogue Resampling Method

While the perfect past-present analogue couplet may not exist, there are some parameters that can be selected to optimise the analogy. To maintain the physical consistency of the ARM, reconstruction we establish the following conditions, largely following Pfister et al. (2020):

1. The identification of suitable analogues depends first and foremost on the definition of the reference period (analogue pool), the library that is searched for the nearest neighbour to the day of interest. Based on data availability, as
discussed above, 1950-2020 was used as the reference period for the reconstruction.

2. Analogue days must have the same synoptic-scale weather conditions – i.e. wind fields – as the day that is to be reconstructed. The pool of possible analogues is thus limited to days that share the same Weather Type (WT) as the target day. Using the daily WT classifications reconstructed by Schwander et al. (2017) the analogue is restricted to the most likely WT plus any additional WTs in order to amount to a combined probability of at least 95%.

3. Analogue days must belong to the same season as the day of interest and are therefore limited to a temporal window of ±30 calendar days around the target day.

4. Finally, the best analogue(s) are chosen based on the smallest 'distance' between all observations; a measure that is calculated based on station data.

According to these points, the best analogue is decided as the day in the reference period with the same weather type, within
the same time window that minimises the difference between certain meteorological variables from a specific set of stations with respect to the target day in the past.

With these conditions set and the data pre-processed, the ARM is applied. Following Pfister et al. (2020), the distances between the historical and reference data are calculated by means of the root mean square error (RMSE):

$$d(\mathbf{x},\mathbf{y}) = \sqrt{\frac{1}{n}\sum_{i=1}^{n}(x_i - y_i)^2} \tag{2}$$

where $\mathbf{x}$ and $\mathbf{y}$ are respectively the vector of observed values and the vector of predictor values and $i$ denotes the observations within these vectors. Accordingly, the smallest distance $d(\mathbf{x},\mathbf{y})$ between the two vectors corresponds to the day that is chosen as best estimate for the given day in the past. With the underlying assumption that errors are unbiased and follow a Gaussian distribution, the RMSE is more reliable if the sample size is large (Chai and Draxler, 2014). RMSE is a useful distance measure because the squared terms ensure that higher weight is given to high errors, meaning larger deviations are
punished more. Once the closest analogue day is found, that day is extracted from the pre-processed E-OBS and ERA5 datasets, the mean seasonality is added back onto the deviation fields. We refer to these resampled fields as ARM reconstructions.



### 2.4   Post-processing

Considering only the single best analogue usually results in a lower skill than if more analogues were considered (Bontron
and Obled, 2005) because one analogue – even if it is the best overall – may not perfectly represent each observation.
Therefore, the temperature and SLP reconstructions from the ARM are further improved. Using a method borrowed from
data assimilation techniques known as Kalman filter (Kalman, 1960) the ARM reconstructions are adjusted to better
represent station observations. We use an off-line variant of the Kalman filter also termed Ensemble Kalman fitting (EnKF,
Bhend et al., 2012; Franke et al., 2017). The ARM reconstructions are taken as the background, which is adjusted based on
the observations. This approach has been used to obtain climate reconstructions that align both with past observations and
the physics used in climate models on a monthly to annual scale. In this study we follow the application of the EnKF adopted
in Pfister et al. (2020).

Using the best analogue as the background (or first guess) and best $n$ analogues as an ensemble, the EnKF will essentially
minimise a least-square errors problem. The vector of the true atmospheric state $\mathbf{x}$ minimises the cost function $J$ according
to:

$$J(\mathbf{x}) = (\mathbf{x} - \mathbf{x^b})^\mathbf{T}(\mathbf{P^b})^{-1}(\mathbf{x} - \mathbf{x^b}) + (\mathbf{y} - H[\mathbf{x}])^\mathbf{T}\mathbf{R}^{-1}(\mathbf{y} - H[\mathbf{x}]) \qquad (3)$$

where $\mathbf{x^b}$ is the background. $\mathbf{P^b}$ is the background error covariance matrix that is calculated from the $n$ best analogues. $\mathbf{y}$
is the vector of station observations and H is the operator that extracts the corresponding grid cell and variable of the
observations from the model space. $\mathbf{R}$ is the covariance error matrix of $\mathbf{y}$-H[$\mathbf{x}$], meaning it contains the errors of both the
observations and the forward operator; it is assumed to be diagonal. The error of the early instrumental observations used
in the reconstruction was estimated following the approach of Wartenburger et al. (2013), which calculates for a given
candidate series from a number of neighbouring observations the daily difference in variance for each pair of stations. We
assume that this difference depends linearly on the squared Eucledian distance between the stations and so a linear
regression can be fitted between variance and squared distance. The average standard deviation of all points to the least
squares fit is taken as the observation error: 2.2 K for temperature and 3 hPa for SLP. The relatively high temperature error
can be attributed to multiple sources, such as the improper shading of the thermometer, inaccurate human reading of the
observations, the measuring of an unrepresentative local climate, or even due to the imprecise assignment of observation
times and daily mean calculation in the data rescue process (WMO, 2008).

A new best estimate of $\mathbf{x^a}$ for the true atmospheric state $\mathbf{x}$ is thus calculated from this Kalman filter based offline data
assimilation:

$$\mathbf{x^a} = \mathbf{x^b} + \mathbf{K}(\mathbf{y} - \mathbf{Hx^b}) \qquad (4)$$

$$\mathbf{K} = \mathbf{P^b}\mathbf{H^T}(\mathbf{R} + \mathbf{HP^b}\mathbf{H^T})^{-1} \qquad (5)$$



Here, $\mathbf{x^a}$ is the updated state vector and H is the Jacobian matrix of $H[\mathbf{x}]$. K is the Kalman gain calculated from the ensemble (shown in Eq. 5), an *n x m* matrix describing the relative weight given to the observations and the current state estimate. Note that mean and anomalies from the mean are updated separately. A more detailed explanation of the implementation the fitting procedure can be found in Bhend et al. (2012) and Pfister et al. (2020).

The ensemble size *n* is set to 50. Using a finite ensemble to approximate the background covariance error leads to spurious correlations, which tends to 'over-correct' and reduce the variance analysis. In other words, the random correlations arise from very distant and uncorrelated locations correlating by chance in the model space and therefore small unphysical updates occur in the assimilation that make the reconstruction over-confident. Spatial localisation is a strategy often implemented to reduce these effects (Houtemaker and Mitchell, 2001; Franke et al., 2017) and can be defined with a function that redefines $\mathbf{P^b}$ such that covariances decay exponentially as a function of distance:

$$\mathbf{P}^{\mathbf{b}}_{i,j} = \frac{1}{n-1} \sum_{k=1}^{n} \mathbf{x}'^{\mathbf{b}}_{i,k} \mathbf{x}'^{\mathbf{b}}_{j,k} exp\left( -\frac{|d_i - d_j|^2}{2L^2} \right) \tag{6}$$

with *n* being the different ensemble members. $|d_i{-}d_j|$ is the distance in km between grid box at position *i* and grid box at *j*, *L* being the cut-off distance. Based on the coverage of historical stations and the size of the study area, the decorrelation distance *L* was chosen to be 750km for temperature and 1500km for SLP.

## 2.5   Evaluation

A leave-one-out validation in time and space is performed on the daily gridded temperature and SLP reconstructions of the winter months NDJF within the reference period 1950-2020. For each day of the reconstructions, the best analogue was calculated with the same requirements as for the winter 1788/9, excluding a ±5-day window around the target day, as spatial patterns of neighbouring days might resemble each other too much and lead to misleading analogues being selected (Pfister et al., 2020). The ARM reconstructions for the reference period were likewise post-processed with the ensemble Kalman fitting.

To assess the quality of the winter 1788/9 reconstruction, this period is validated against weather observations that were not used in the reconstruction. Two independent stations were selected, namely Zurich in Switzerland (Pfister et al., 2020) and Cadiz in Spain (Barriendos et al., 2002). For the comparison, the corresponding locations of Zurich and Cadiz are extracted from the gridded reconstructions of the ARM and EnKF.

The reconstructions are assessed for their quality following metrics of skill and reliability commonly used in the validation of field forecasts (Wilks, 2019). All measures were calculated on anomalies from the mean seasonal cycle such as not to confound skill that comes from seasonality. The Pearson correlation is used as a first performance metric. The RMSE is used to assess the error magnitudes. The validation also assesses the systematic bias between the E-OBS (temperature) and ERA5 (SLP) datasets and the respective reconstructions.

Additionally, the mean squared error skill score (MSESS) is used to evaluate the reconstructions' predictive skill, according to the following equation:


$$MSESS = 1 - \frac{\sum\limits_{i=1}^{n}(x_i^{rec} - x_i^{ref})^2}{\sum\limits_{i=1}^{n}(x_i^{clim} - x_i^{ref})^2} = 1 - \frac{MSE_{rec}}{MSE_{clim}} \tag{7}$$

The MSESS is a function of the reconstruction $x^{rec}$, the reference data $x^{ref}$ from E-OBS/ERA5, and a perfect or "no-knowledge" prediction $x^{clim}$, in this case the mean climatology. For the validation in time $i$ denotes the time step and for the validation in space it refers to the grid cell. This dimensionless skill score is positive (negative) when the accuracy of the reconstruction is greater (less) than the accuracy of the climatological prediction (Murphy, 1988). The maximum value the MSESS can have is 1, which equals a perfect predictive skill; negative values imply the reconstruction's predictive skill is worse than that of the 305 climatology and can range to minus infinity; an MSESS value of 0 indicates $MSE_{rec} = MSE_{clim}$ and implies no skill (Jolliffe and Stephenson, 2012).

### 2.6 Uncertainty and limitations

The uncertainties begin with the observations and their pre-processing. As mentioned in the previous Section, historical observations are subject (to differing degrees) to a number of potential inaccuracies and biases. Gridded datasets derived 310 through the interpolation of modern station data may also contain errors. These inaccuracies may be introduced due to the propagation of errors in the underlying station data; in the case of the E-OBS dataset, not all stations used for interpolation have been homogenised, leading to imprecise estimations of absolute values and variance of grid cells (Hofstra et al., 2009).

The main limitation of the ARM is the size of the analogue pool (in this study 71 years, 1950-2020). Given the large variety of atmospheric flow, the shorter the reference period, the more unlikely it is to find two matching atmospheric states (Van 315 Den Dool, 1994). If the observational record from which analogues are sampled is too short, the distances between these matches could be too large for them to be strictly called analogues. Indeed, the criterion for what constitutes a truly good analogue is arbitrary and depends heavily on the specific methodological choices employed (see Sect. 2.3). Due to their low probability, the size of the analogue pool is an even more pertinent issue for extreme events as they are by definition rare, meaning they do not occur often enough for there to be 'worthy' analogues (Flückiger et al., 2017; Pfister et al., 2020). 320 However, the EnKF approach partly corrects for this.

Another difficulty that affects the ARM is the spatial coverage of the historical station data, which serves as input for the reconstructions. The station coverage in Figure 1 showed gaps in the Iberian Peninsula, South-Eastern and Eastern Europe, and Scandinavia, where the lack of station density might negatively impact the skill of the predictions in these regions, especially for temperature, which has a higher spatial variability. That said, inaccuracies in the data cannot be ignored when 325 reconstructing pre-industrial climate and weather. A final source of error specific to our study are the uncertainties due to the subtraction of the intercentennial temperature trend from the reference data discussed in Sect. 2.2.



## 2.7    Analysis

Following the evaluation of the reconstructions, in Sect. 3.4 we turn to look at the reconstructions themselves and study the

winter 1788/9 in more detail. First of all, we provide some climatological context to the winter 1788/9 with a literature analysis. We show what is known about this event from the point of view of instrumental records and its impact on human activities by exploring the existing literature on the winter 1788/9. The maps from our reconstruction can then be used to complement and enhance the current state of knowledge about the event; we demonstrate this on behalf of an example with the SLP maps.

The spatial information from the reconstructed fields of this winter allow for comparisons to be made with other extreme cold winters for which we already have spatial information from E-OBS dataset. Such a comparative exercise might help us better situate the winter 1788/9 as an extreme – how it is alike or differs from other comparable extremes. We illustrate this by comparing the reconstructed temperature fields of the winter 1788/9 with three of the coldest winters in the second half of the 20th century: 1955/6 (Dizerens et al., 2017), 1962/3 and 1984/5. The choice is based on the study by Twardosz et al.

(2016), which identifies severe winters with the most extreme cold months based on 60 weather stations across Europe in the period 1950-2010. For each winter we then calculated the cold spell index eca_cwfi (Schulzweida, 2019), i.e. the cold-spell days index for each winter (NDJF) with reference to the 10th percentile of the daily mean temperatures for the period 1950-2020 (NDJF) . In other words, for each grid cell counted is the number of days where, in intervals of at least 6 days, the daily winter temperatures are below the 10th percentile of the reference period. In a next step, we identified cold spells

below the 1st percentile of the reference period 1950-2020 using the *detect_event()* function from the *heatwaveR* software (Schlegel and Smit, 2018). The time series used as input are calculated from the daily field mean of a large area around Central Europe, ranging from 2-25° E and 44-55° N.

## 3    Results and discussion

As described in Sect. 2.5, we performed a leave-one-out validation for the winter months NDJF in reference period 1950-

2020 to assess the skill of the reconstruction. These reconstructions, both ARM and the post-processed EnKF, are compared to the gridded datasets E-OBS (temperature) and ERA5 (SLP). The validation against independent observations tests the skill of the reconstruction specifically for the winter 1788/9. This section presents the results of the validation and explores its implications. The aim is thus to assess whether the analogue resampling method managed to capture the temporal evolution and spatial patterns of over Europe given the set of station observations that were used, as well as show the extent to which

the ensemble Kalman fitting improved the accuracy of the reconstructions. In a final subsection we return to the reconstructed event itself, in two parts: firstly, a literature analysis investigates some of the weather patterns that characterised the winter 1788/9 and the social crises that ensued; secondly, a reconstruction analysis looks at some of the reconstructed maps, then compares the winter 1788/9 to three cold winters of the modern period, and finally suggests ideas for further work on the topic.



### 3.1 Reconstruction

#### 3.1.1    Leave-one-out validation in time

Figure 2 shows the validation over time for temperature reconstructions. The top row of panels reveals validation metrics of the best analogue from the ensemble (ARM), while the bottom row depicts the same for the post-processed reconstruction (EnKF). The best analogue already shows a significant correlation with an average coefficient of 0.68, with higher values between 0.7 and 0.8 concentrated in the region North of the Alps, extending from western France and England to Poland (a). The post-processing improves the correlation to a mean coefficient of 0.82 over the study area, reaching values above 0.9 for a large strip of land stretching from the Pyrenees to North-East Europe (e). Looking at the RMSE (b, f), we see that the best analogue has errors up to 3 °C in Central and Western Europe and from 3 to 8 °C in Eastern Europe, the latter of which is improved in the postprocessing, mostly in regions with station coverage. The mean bias over the winter months NDJF for the ARM (c) is positive in the North-East of the study area and and slightly negative in the South-West, a pattern that is largely corrected in the EnKF (g), save for the introduction of a warm bias in central Germany due to an overestimation in the Erfurt reference series. The reconstruction generally shows good skill in replicating the temporal succession of temperature, with an average MSESS of 0.4 in the ARM (d) and 0.65 with high values above 0.9 for the EnKF (h). Overall, the post-processing has improved the accuracy of the reconstruction, however, the extent of the improvement is limited to areas with station data. No temperature observations were used in the Iberian peninsula, for instance, hence there is only poor improvement on an already poorly reconstructed region.

The validation over time for SLP shown in Figure 3 offers a more promising picture. The Pearson correlation coefficient for the ARM are higher than 0.6 over most of the study area (a), which is significantly improved in the EnKF with coefficients higher than 0.9 over most of Western, Central, and Northern Europe (e). Over these regions the EnKF manages to reduce the RMSE by 1 to 3 hPa, but it shows little or no improvement over areas where there are no station observations. Results from the South-East of the domain are not reliable and in the case of Russia should be interpreted with caution. The mean bias (c, g) is reduced by the post-processing leaving only some areas with a negative bias around -0.3 hPa. The MSESS (d, h) shows a similar pattern to the correlation, indicating an excellent skill over most of Europe in the EnKF, which rapidly deteriorates towards the East. Thus, we can confidently say that the post-processed SLP reconstruction can accurately replicate the temporal succession of SLP fields over most of Central, Western, and Northern Europe. The extent of the improvement of the EnKF compared to the ARM is greater for SLP than for temperature, due to the overall lower spatial variability of SLP and the consequent selection of a long decorrelation distance $L$ for the assimilation.

#### 3.1.2    Leave-one-out validation in space

The reconstructions are further assessed for their ability to reproduce spatial patterns. To this end, temperature and SLP reconstructions are validated based on quantiles of spatial mean temperature and SLP anomalies, calculated respectively from the E-OBS and ERA5 datasets for each day in the period 1950-2020 (NDJF). The spatial validation of the post-processed temperature reconstruction (Fig. 4, top) shows a considerable improvement relative to the ARM both in median and spread



of values. Spatial correlations do not differ significantly across quantile groups with median values around 0.75. The RMSE indicates that error magnitudes increase as values approach the lower percentiles, i.e. infrequent temperature extremes.

The ARM temperature reconstruction tends to overestimate cold extremes, which have a larger spread of biased values with a median of about 1.3 °C, though this is improved by almost 1 °C in the EnKF; nevertheless, uncertainties remain large, shown by the whiskers extending beyond 2 °C in both directions. Larger errors and biases with extreme negative values can be explained by the lack of suitable analogues for cold extremes given their lower occurrence. That said, the areas of extreme cold are well-represented in the EnKF, showing a high MSESS below the 10% with the entire Inter-Quartile Range (IQR) above

0.75. On the other hand, the spatial pattern of more common winter temperatures is harder to predict with accuracy, though the MSESS still indicates a good predictive skill compared to the climatology.

Meanwhile, the spatial validation of the post-processed SLP reconstruction (Fig. 4, bottom) shows an even greater improvement relative to the ARM compared to temperature. This is especially the case for error magnitudes: RMSE medians were improved on average by 3 hPa, yet the long upper whiskers of the boxplots indicate that some grid cells have errors

between 6 and 8 hPa. These values are likely due to the absence of observations in areas of the domain such as Turkey, the Dead Sea, and Russia. Correlations improve considerably with the Kalman fitting, as does the mean bias, indicating a more balanced spatial pattern. In the ARM, low SLP extremes tend to be overestimated and high SLP extremes are underestimated, hinting again to the fact that there are likely insufficient analogues from which to draw extremes; the EnKF managed to minimise this effect as well as reduce the spread of biased values. MSESS values from the ARM show better skill for lower

and upper quantiles and worsen approaching the median. This pattern is no surprise as days around the median are closer to the average climatology, hence the denominator in Eq. 7 becomes smaller, leading to a lower MSESS. Once again, the EnKF manages to significantly improve the predictive skill of the reconstruction: the median of all quantile groups have a MSESS above 0.75.

Spatial patterns of temperature and SLP over Europe are physically consistent and generally well-represented in the EnKF

reconstructions, which display a substantial decrease in the spread of values across all metrics compared to the best analogue alone. Nevertheless, reconstructions of 18th-century weather maps depend on the quality of the early instrumental meteorological observations that are used as input, which are typically more inconsistent and have larger measurement errors relative to modern-day observations. The next section explores this point.

### 3.1.3 Validation against independent observations

Reconstructed temperature and SLP were compared to two independent measurement series for the winter 1788/9 from the stations Zurich and Cadiz. These locations represent areas with respectively high station coverage and poor to no station coverage. Unlike the leave-one-out validation, the validation against independent observations was performed on absolute values rather than anomalies. Figure 5 illustrates this comparison in the form of a time series (top), a scatter plot (centre), and a Quantile-Quantile plot (bottom). Here too, for the most part, we see that the EnKF improves the overall accuracy of

the reconstruction compared to the observations. Compared to the Zurich temperature observations, the ARM alone manages to capture the temporal succession of temperature over the four winter months and exhibits a correlation of 0.82





with the independent observations; in the EnKF this linear relationship is slightly improved to 0.89 in the EnKF. The RMSE also shows an improvement from 3.4 to 2.7 °C, as does the mean bias from 0.67 to -0.13. The spread of values around the 1:1 line is reduced, however, the EnKF slightly worsens the underestimation of cold extremes, as can be seen in the first week

of January, likely due to the high altitude neighbouring stations such at Gotthard and Hohenpeissenberg registering colder temperatures in these days. Some of these disparities can be linked to differences between local measurements and the spatially coarser gridded data. The difficulty in accurately estimating cold extremes also stems from the lack of available analogues in the 71-year analogue pool.

In data sparse regions such as southern Spain, the EnKF shows no significant improvement on a weak best analogue.

Comparisons with the Cadiz temperature series show a large spread of values and poor correlations for the ARM; the assimilation did not significantly improve the reconstruction in this area. The linear relationship is 0.5 for the ARM and 0.58 for the EnKF, which is around the expected value for this region from the validation over time; the RMSE remains unchanged at 3.9 °C, and the mean bias worsens from -2.6 to -3 °C. The reconstructed winter at Cadiz is substantially colder than the observations show. That said, doubts about the quality of the Cadiz temperature series cloud the validation: not only are

these daily means based on observations taken only once a day at noon, but there are several gaps in the series. Moreover, these measurements predate the reorganisation of the facilities and equipment of the observatory that began after 1789 (Barriendos et al., 2002). Thus, it could well be that the instrument itself was poorly exposed and the observations may not capture the full extent of the cooling.

SLP reconstructions show better results in both observation series. The ARM manages to capture the general development

of SLP in Zurich during the winter 1788/9, exhibiting a strong correlation with the observations of 0.93. The EnKF improves this to 0.98, while the RMSE is lowered from 4 to 2.5 hPa, though the mean bias is marginally increased from 1.4 to 1.5 hPa. Notably, the EnKF improves the accuracy of extreme values. This happens for Cadiz as well, though to a lesser degree. Here, the spread of values around the 1:1 line is larger, though the mean bias averages out to -0.3 hPa and remains unchanged in the EnKF. The linear relationship between reconstruction and observations is shows significant improvement from an R of

0.55 to 0.8. The RMSE is reduced from 7 to 4.8 hPa. It is worth keeping in mind that the RMSE punishes larger deviations more heavily. Overall, there is a slight overestimation of low SLP values and an underestimation of high SLP.

The validation against independent observations has shown that the post-processed reconstructions manage to accurately reproduce the day-to-day variability of SLP during the winter 1788/9. The accuracy of the reconstructed temperature fields is good in locations with dense station coverage and poor in regions far away from any assimilated station data. The extent

of the cold extremes is hard to validate as station temperatures are highly variable to due local influences and may not be captured by the gridded data.

Overall, the reconstructions show satisfying results: the validation for the EnKF temperature reconstruction shows good correlation and skill score, though only in regions with decent station coverage, whereas the results for SLP look even more promising with even better validation metrics. The results prove that Pfister et al.'s (2020) suggestion to extend the method

further back in time, to a larger study area, and possibly to other variables, was largely successful. That said, certain aspects



of the results stress the need for caution. Compared to their study, our post-processed temperature reconstruction shows significantly higher errors; this can be partly attributed to higher measurement uncertainties in our dataset and the uneven station coverage over a much larger domain with strongly varying topography. Furthermore, our results show that accurately reconstructing extreme values is challenging and relies on a long enough analogue pool from which to select suitable

matches. Furthermore, Pfister et al. (2020) found a better skill for summer reconstructions than for winter ones: it is very possible that reconstructing the summer 1789 would yield lower errors.

Any shortcomings revealed by the validation are mainly the expected ones described in Sect. 2.6: successfully capturing extremes depends on having a large enough analogue pool that contains rare analogues; and poor station coverage (and quality) leads to omitted information and thus to poor correlations, higher errors, and lower skill scores. In short, the

limitations of the ARM are defined by the available data. To this list we may add the choice to reconstruct a relatively large spatial domain. Indeed, Ruosteenoja (1988) and Van Den Dool (1994) have argued for the existence of a three-way relationship between the length of the meteorological archive, the size of the spatial domain (or degrees of freedom), and the resulting quality of the analogy: errors increase with a bigger domain, yet decrease with a longer reference period archive.

The relative paucity of historical data points in the domain highlights the importance of further data rescue efforts,

especially the digitisation and quality control of early instrumental observations. The demanding time resources used for gathering the needed observations for this study point to the pressing need for collaboration across different institutions to rescue, correct and store daily measurements in common open access repositories, which would facilitate and accelerate the pace of further studies on the subject.

### 3.2    The winter 1788/9

### 3.2.1    Literature analysis

Europe's winter of 1788/9 takes place towards the later stages of the Little Ice Age, towards the end of a century that had seen many climatic anomalies, among which were numerous cold winters. Across the globe, in fact, the decade of the 1780s was marked by increased climate variability containing a number of outstanding temperature and rainfall extremes (Damodaran et al., 2018). These included protracted negative phases of the North Atlantic Oscillation (NAO), the effects of

the Laki volcanic eruptions in 1783 (Brázdil et al., 2010), as well as particularly extreme episodes of El Niño 1782-1784 and La Niña 1785-1790, which had severe effects on the cycle of floods and droughts that can be linked to the famines in India and Japan during these years. As Kington (2010) remarked, the late 18th century climate had "entered a less stable mode, owing to a complex interactions of various factors." Using instrumental and documentary data, Barriendos and Llasat (2003) identified a period of anomalous hydrometeorological activity in the Western Mediterranean known as the 'Maldá Anomaly';

particularly during 1780 and 1795 this region saw an increase in the frequency of exceptionally dry and wet extremes, including some catastrophic snowfalls. In accordance with this characterisation of the climate, Kington (2010) argues that the decades after 1780 were marked by a southward displacement of the warming waters of the North Atlantic Current and a corresponding shift of the prevailing westerlies to lower latitudes, giving way to more anticyclonic situations over northern Europe.



The years around the winter 1788/9 even show pronounced seasonal variability. After a wet autumn on the Iberian
Peninsula and a rather mild winter 1787/8, spring 1788 had been one of the driest on record for England and parts of France;
summer 1788 had been hot and dry, dotted with some heavy thunderstorms, including the devastating hailstorm of 13 July.
The cold spell that would turn out to be one of the harshest winters on record started in late November and lasted until mid-
January. During this "cruel winter" the Venice Lagoon froze over, as well as other lakes and rivers across Central Europe

(Societas Meteorologica Palatina, 1791; Gisler, 1985; Camuffo, 1987); in London, a frost fair was held on the Thames for the
first time since the severe winter of 1739/40 (Kington, 1980). By the end of January, the worst of the intense cold was over;
milder weather in February led to the thawing of frozen rivers and melting of large amounts of snow that had accumulated
in several regions. By spring 1789, there had been numerous floods that devastated thousands of hectares of farmland
(Barriendos et al., 2000; Fagan, 2001; Brázdil et al., 2003; Kington, 2010).

Observers at the time were aware of the outstanding severity of the winter 1788/9 (Cotte, 1789; Strnadt, 1793). Their
weather observations allow us to see the progression of the cold spell from November to February (Fig. 6). By 20 November,
most stations in Europe show temperatures plunge below zero and remaining bitterly cold until the second week of January,
except for a brief interval of mild stormy weather over Christmas (Pfister et al., 2019). The cold outbreaks coincide with a
breakdown of the zonal flow over Europe, interrupting the usual poleward pressure gradient and its associated warm moist

air advection. Indeed, the CAP7 Weather Type Classifications by Schwander et al. (2017) show that days during this season
were mostly of type 6 ("North"), along with some of type 1 ("North East, indifferent") and type 4 ("East, indifferent"), pointing
to the flow of cold continental air towards Central Europe. Corroborating this picture, pressure series from southern Europe
have shown repeated low pressure situations in the Mediterranean (Barriendos et al., 2000). This is not surprising seeing as
cold extremes in Europe are known to be associated with specific winter atmospheric anomalies that are linked to a negative

phase of the Northern Atlantic Oscillation (NAO) (Cattiaux et al., 2010; Sillmann et al., 2011; Pfahl et al., 2014). A NAO-pattern
is associated with an enhanced northerly and easterly wind and persistent atmospheric blocking occurrences in northern
Europe, which induce strong cold advection of polar air masses into lower latitudes (Takaya and Nakamura, 2005; Buehler et
al., 2011). The reconstructions in this study would confirm the spatial extent and daily evolution of this pattern.

        Lastly, the winter 1788/9 has value not only climatologically but also historically for the social crises it aggravated and
those it engendered. The extreme weather fluctuations in this period had created instability in the marketplace, which
harvests alternate abruptly between plentiful and scarce (Fagan, 2001). The prolonged droughts in 1788 had led to crop
failures, which soon led to high grain prices and food shortages, resulting in widespread hoarding, thieving and much human
suffering. The ensuing cold winter could not have come at a worse time. Major rivers froze over and heavy snowfall blocked
roads, bringing most commerce to a halt and preventing the import of grain. The spring floods exacerbated the subsistence

crisis even further. Bread riots broke out in Flanders, France, and elsewhere. In Barcelona, an upheaval known as the
"Rebomboris del Pa" rioters looted and set fire to the municipal supplies and ovens (Barriendos et al., 2000). France had been
dealing with a deteriorating social and political state of affairs for decades and the adverse weather completely overwhelmed





the capacity of the authorities to contain the situation. The winter 1788/9 may have hastened the start of an inevitable outcome that culminated in the French Revolution, which broke out the following summer.

### 3.2.2 Reconstruction analysis

The encouraging results from the evaluation of our reconstructions hold promise for research on high-resolution historical climate reconstructions and the interpretation of past extreme weather and climate variability. More specifically, our reconstructions now allow a richer discussion about some aspects of the winter 1788/9 that were previously unexplored. For instance, the maps allow a more detailed spatial assessment of the synoptic patterns over Europe during this period, something which observations alone cannot provide. Figure 7 shows daily reconstructed EnKF maps of SLP from 21 December to 5 January. These correspond to weeks during the winter with marked swings in the zonality index, which brought about brief but pronounced warmings around Christmas and the New Year, each one followed by sudden, sharp coolings (see Fig. 6). 24-25 December display a pronounced dipole pattern with a strong low pressure centred over North-Eastern Europe; one can easily imagine a storm moving in from the Atlantic blowing milder temperatures towards Europe. Soon after a high pressure emerges in the West and by 28 December, it has settled over much of the North, advecting cold continental air from the North-East. By 1-2 January, the high pressure has weakened giving way to lower SLP values to move across the land, only to be replaced by reinvigorated blocking situations in the following days. By 5 January, an intense high pressure system bringing freezing air into Central Europe extends over the continent, with centre over the North ranging from England over Scandinavia to southern Finland, with values above 1045 hPa. According to the spatial validation, these SLP spatial patterns are reconstructed fairly accurately.

### 3.2.3 Comparison with more recent cold episodes

This winter can now be compared to other episodes of extreme cold to identify important differences or similarities in spatial extent, timing, frequency, and intensity. Applying the eca_cwfi index (Schulzweida, 2019) to each winter illustrates how each extreme is quite unique in the number and spatial distribution of cold-spell days (Fig. 8). The winters 1788/9 and 1962/3 have long cold-spells reaching into Western Europe, 1984/5 was particularly severe in Finland and the East, whereas 1955/6 is more homogeneously distributed across the land. The winter 1962/3 has decidedly the most days under the 10th percentile of the reference period with more than 30 cold-spell days in much of Central and Western Europe, and more than 50 in England and Belgium; however, it is also the winter with the least amount of anomalous cold days in Northern Europe.

While the eca_cwfi index gives an idea of the regions where the winters were more intense, it is based on a 4-monthly average and does not distinguish between several different spells within the winter, nor does it say much about the intensity of the extremes itself. Looking at the 1st percentile cold spells averaged over Central Europe makes for a more distinct characterisation of these winters. The approach of Hobday et al. (2016) to event detection as implemented in the R command *detect_event()* identified 3 events for 1788/9 and 1962/3 and only 1 for 1955/6 and 1984/5: Table 2 summarises the relevant statistics for each extreme cold spell. One aspect that immediately sets the winter 1788/9 apart from the others is its timing; it is the only winter with a cold spell as early as November. Most of the late 20th-century winters, in fact, have cold spells



occurring in January and sometimes lasting well into February. Out of the 8 listed cold spells, the one from 1788-12-27 to 1789-01-09 has the highest mean and maximum intensity relative to the seasonality, respectively with -13.4 °C and -16.5 °C. The 1788-12-13/1788-12-20 has the second highest mean intensity at -11.53 °C. It is not surprising that December 1788 ranks amongst the coldest in Central Europe for the last 300 years. Interestingly, 1984/5 stands out for the longest cold spell with

as many as 27 days from 01-31 to 02-26, though the rate of onset and decline of the event are lower. The second December cold spell in 1788 has an onset rate as high as -2.8 °C/day, meaning it was an abrupt cold wave that reached its peak intensity extremely fast, as confirmed by the sudden persistent blocking seen in Figure 7. From this we can conclude that the winter 1788/9 in Central Europe was an early one, marked by 3 extreme cold spells that increased progressively in duration and intensity, with moderate to high rates of onset and decline.

The reconstructions nicely complement and, more importantly, add layers to the literature covered in Sect. 3.4.1. Even more could be learned comparatively about the winter 1788/9 if these reconstructions were extended to other extreme winters containing severe cold snaps in the pre-industrial period, such as 1708/9, 1775/6, 1783/4 or 1819/20. Other seasons of outstanding (or even non-extreme) weather and climate could also be reconstructed and analysed: as previously hinted, these might in fact yield lower errors. The ARM method has shown to be suitable for reconstructing SLP and temperature

fields over Europe, although for temperature, a vast spatial domain and poor station coverage have shown to be a significant hurdle. For temperature it could be worth focusing on smaller regions and aim additional efforts at the collection of corrected early instrumental observations and ensuring extensive station coverage.

**Table 2:** Detected cold spells for the winters 1788/9, 1955/6, 1962/3, and 1984/5 below the 1st percentile of mean daily temperature in the period 1950-2020 all averaged over Central Europe (2-25° E, 44-55° N). Shown is the duration of the cold spell in days, its start, peak
and end dates; for each event are also shown the mean and maximum intensity [°C] relative to the seasonal climatology as well as the rate of onset and decline of the event [°C/day].

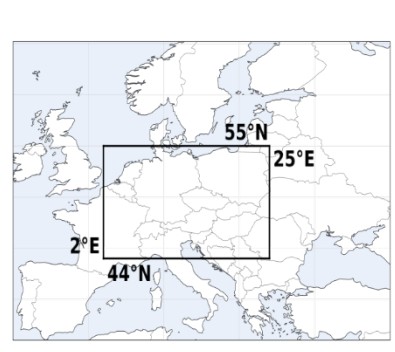

| duration | date | | | intensity [°C] | | rate [°C / day] | |
|---|---|---|---|---|---|---|---|
| days | start | peak | end | mean | max | onset | decline |
| 5 | 1788-11-24 | 1788-11-26 | 1788-11-28 | -8.3 | -10.1 | -1.68 | -1.67 |
| 8 | 1788-12-13 | 1788-12-18 | 1788-12-20 | -11.53 | -13.92 | -1.28 | -2.13 |
| 14 | 1788-12-27 | 1788-12-29 | 1789-01-09 | -13.44 | -16.51 | -2.8 | -0.74 |
| 27 | 1956-01-31 | 1956-02-10 | 1956-02-26 | -10.71 | -15.78 | -0.61 | -0.53 |
| 7 | 1962-12-22 | 1962-12-23 | 1962-12-28 | -9.21 | -11.04 | -2.47 | -0.78 |
| 8 | 1963-01-12 | 1963-01-18 | 1963-01-19 | -10.66 | -12.76 | -0.7 | -2.14 |
| 7 | 1963-01-29 | 1963-01-30 | 1963-02-04 | -10.16 | -11.23 | -1.86 | -0.57 |
| 13 | 1985-01-05 | 1985-01-08 | 1985-01-17 | -10.96 | -14.48 | -1.33 | -0.69 |





## 4    Conclusions

We set out to apply the analogue resampling method together with an ensemble Kalman filter technique to reconstruct daily fields of temperature and SLP over Europe for the extreme winter 1788/9. The CAP7 weather types were used as a basis for physical consistency in the analogue search and the spatial information for the fields was taken from the E-OBS and ERA5 datasets, for temperature and SLP respectively. A vital part of the study involved the collection and quality control of numerous early instrumental series from a total of 53 stations. Our final aim was to be able, on the basis of both the applied method and the new reconstructions, to gain more complete daily and spatial information about the extreme winter to expand on the existing literature.

For both variables the EnKF reconstructions represented a substantial improvement on the ARM's best analogue, especially with regard to the mean bias. The temperature validation showed robust results, especially considering the relatively high measurement errors. Station-dense regions such as Western and Central Europe, as well as the North-East, showed good correlations and high skill scores, while station-sparse areas showed poor results and higher errors – a known limitation of the method. The SLP reconstructions were also affected by this pattern but to a much lesser degree; in fact, the validation showed a very high predictive skill over most of the continent. With these results, we proved the suitability of the used method in reconstructing daily temperature and SLP fields. That said, measurement uncertainties, data quality and coverage, and the size of the analogue pool are all important factors that can undermine the analogue approach.

The high spatial and temporal resolution of the such gridded datasets presents clear advantages for the study of extreme weather variability; they can provide realistic circulation patterns and help to locate both the regional extent and intensity of anomalies. Indeed, the analogue approach may prove useful for the reconstruction of other pre-industrial extremes, both cold and warm, thereby helping not only to better understand the events themselves but also to put more recent observed extremes in the context of a much longer period. Comparisons with other extreme winters allowed us to contextualise the reconstructed 1788/9 event and identify characteristics that distinguish it. The winter freeze started early and was marked by intense blocking systems that extended over large parts of Northern Europe; the cold air outbreaks into Central Europe were relatively abrupt and intense, despite being interrupted by brief spells of mild temperature.

To avoid monocausal and determinist explanations, this study has purposely avoided drawing a line between the extreme winter 1788/9 and the events that led to the French Revolution in the summer of 1789 – an undoubtedly significant turning point in European history. Nevertheless, the weather information uncovered in these reconstructions could be useful to historians to study social developments that might have depended on day-to-day weather, or indeed on health and agriculture. More specifically, one could trace the daily impact of weather during this period on political and economic affairs, military operations, or any travels and expeditions. Daily historical reconstructions of the kind presented here have much to offer both climatological and historical research.

*Code and data availability.* The data and relevant code related to this article are available as supplement to this paper. The instrumental series are stored in SEF format, the code as R scripts, and the reconstructed maps in NetCDF format.



*Author contributions.* DP gathered the data, performed the reconstructions, created the figures, and drafted the manuscript. YB provided guidance processing of early instrumental observations. MB, SJ, and RP contributed a number of digitised series and, together with CR, they provided critical feedback to improve the final version of the manuscript. NI helped develop the scripts for the reconstruction. SB helped shape the research and supervised the project.

*Competing interests.* The authors declare that they have no conflict of interest.

*Acknowledgements.* This work was supported by the Swiss National Science Foundation (project WeaR 188701), by the European Research Council (ERC) und the European Union's Horizon 2020 research and innovation programme grant agreement No 787574 (PALAEO-RA), and by the project "Long Meteorological Series" of the Federal Office of Meteorology and Climatology MeteoSwiss in the framework of GCOS Switzerland.. Also for this we acknowledge the help of MeteoSchweiz, the Met Office National Meteorological Archive, the

Universitätsbibliothek Rostock, the Deutscher Wetterdienst, and the Polish Institute of Meteorology and Water Management in providing some of the data. The work of MB was supported by the Catalan Meteorological Office. The work of RP was conducted within the NCN project (grant no. DEC-2020/37/B/ST10/00710). We are grateful to Rob Allan, Ed Hawkins, Jari Holopainen, Heli Huhtamaa, and Andrea Kiss for their help in providing some of the digitised historical observations, scanned sources and metadata. Assistance provided by Lucas Pfister for his design of the original R scripts is much appreciated. We would also like to acknowledge the ERA5 reanalysis and E-OBS datasets, as

well as the data providers in the ECAD project.

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



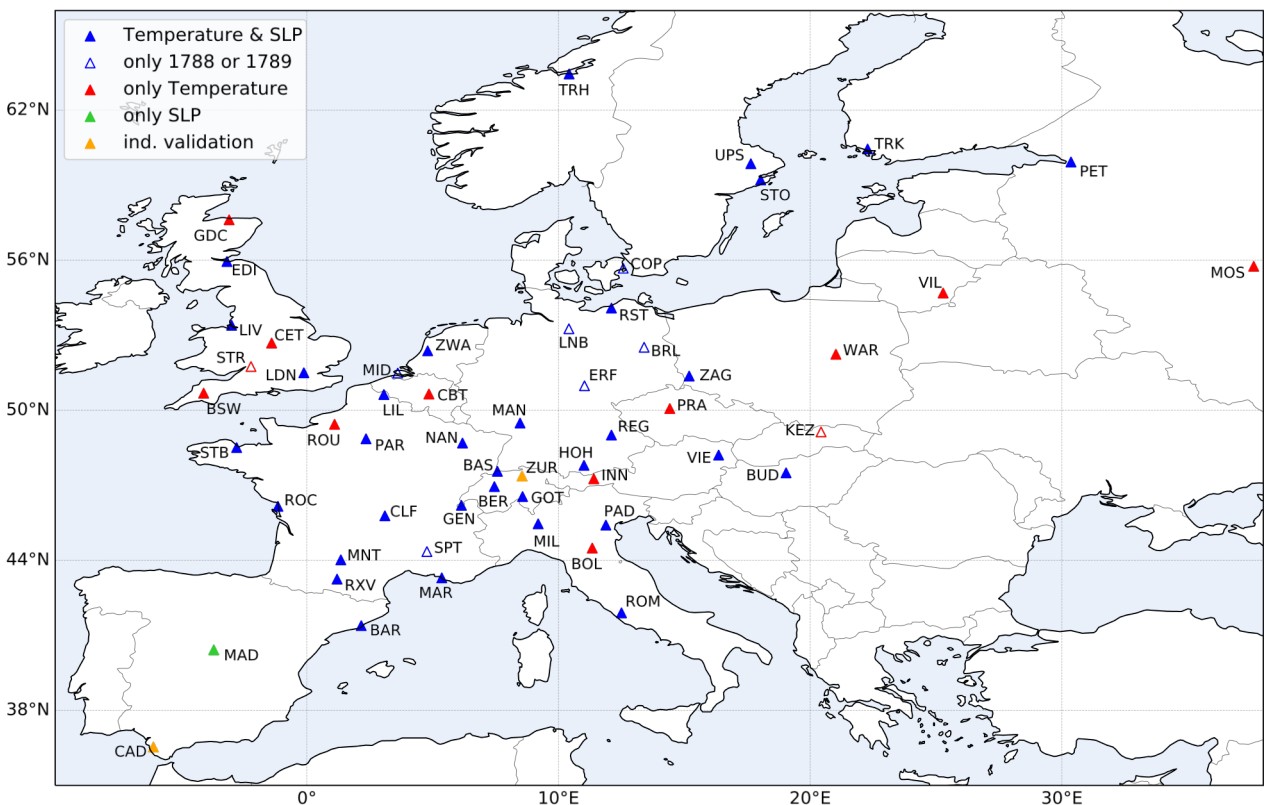

**Figure 1.** Map of stations included in the reconstruction. Distinction is made between stations with series with both temperature and SLP (blue), or just one of these variables (red and green, respectively); shown are also those stations with data for either just Nov-Dec 1788 or Jan-Feb 1789, as well as stations used for an independent validation (orange).






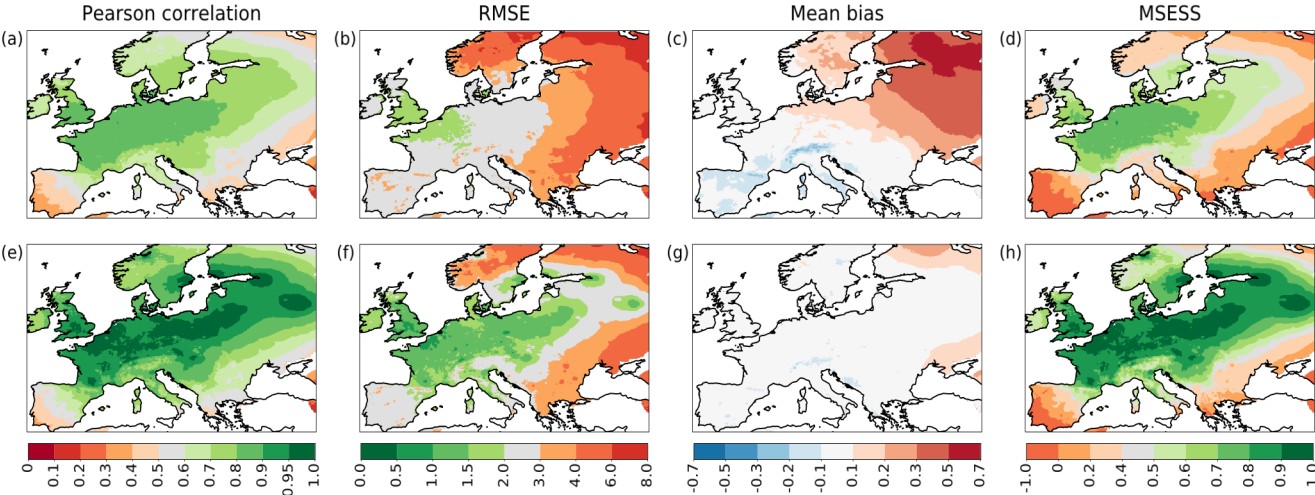

**Figure 2.** Validation of temperature over time for winter months NDJF in the period 1950-2020 for the ARM (a-d) and EnKF (e-h) reconstructions. The reconstructions are compared vertically by validation metrics: shown are the Pearson correlation coefficient (a, e), RMSE (°C) (b, f), mean bias (°C) (c, g), and MSESS (d, h).


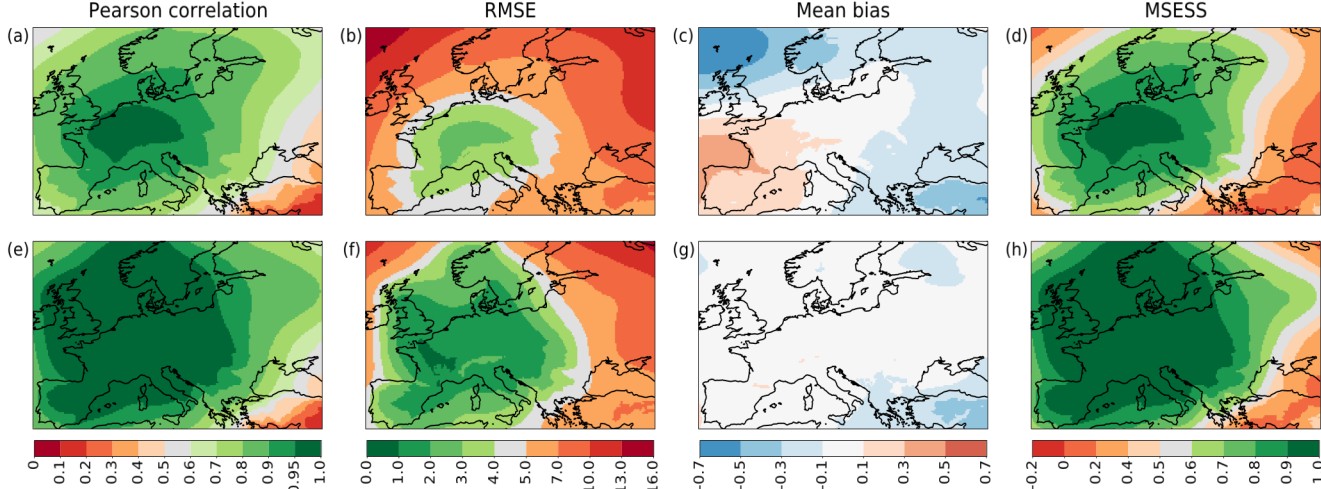

**Figure 3.** Validation of sea level pressure over time for winter months NDJF in the period 1950-2020 for the ARM (a-d) and EnKF (eh) reconstructions. The reconstructions are compared vertically by validation metrics: shown are the Pearson correlation coefficient (a, e), RMSE (hPa) (b, f), mean bias (hPa) (c, g), and MSESS (d, h).






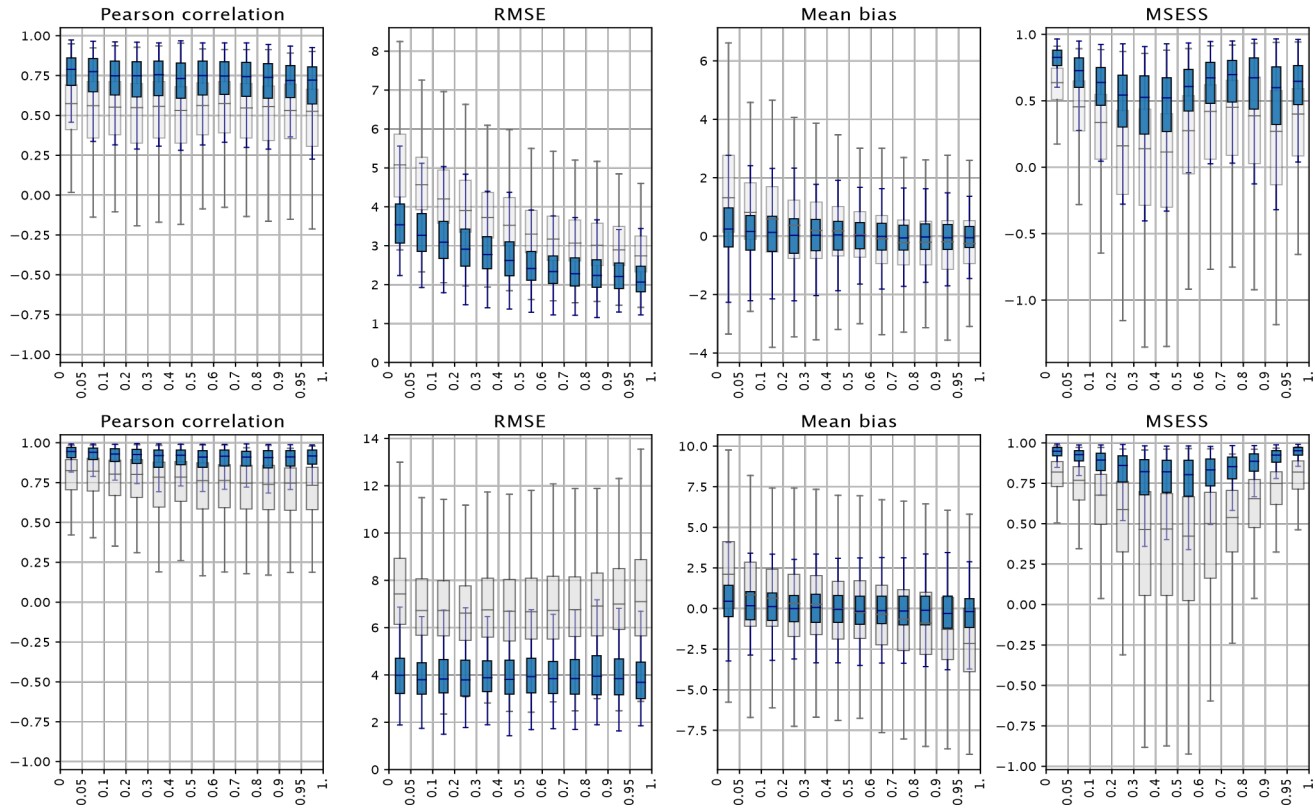

**Figure 4.** Validation over space by mean temperature quantiles (top) SLP quantiles (bottom) for the period 1950-2020 (NDJF). Shown are the results for the ARM (gray) and the EnKF (blue) and their respective metrics: Pearson correlation coefficient, RMSE, mean bias, and MSESS. The boxes represent the interquantile range (75th - 25th percentiles) and the whiskers extend to a distance of 1.5*IQR outside the

boxes.



**Figure 5.** Validation of ARM and EnKF reconstructions of the winter 1788/89 against independent observations from Zurich (**left**) and Cadiz (**right**). The top panels shows time series of temperature and pressure comparing the ARM (gray), the EnKF (black), and observations (coloured). The panels in the centre are plots of the reconstructions (ARM, gray dots; EnKF, black dots) against the observations; the top left corner of the middle panels show validation metrics for the reconstructions, shown are the pearson correlation coefficient, RMSE, and bias. The bottom panels are Q-Q plots displaying the same.





**Figure 6.** Daily mean air temperature from November to February 1788/9 at Rostock, Zwanenburg, Warsaw, Prague, Mannheim, Paris,
Innsbruck, Bern, Clermont-Ferrand, and Milan (top) and a Zonality Index (bottom), calculated as the standardised difference between the
average SLP at Rome-Barcelona and Trondheim-Edinburgh; shown is also a 4-day moving average (red).



**Figure 7.** Daily mean sea level pressure fields over Europe (hPa) from the EnKF reconstruction covering the days from 21 December 1788 to 5 January 1789.



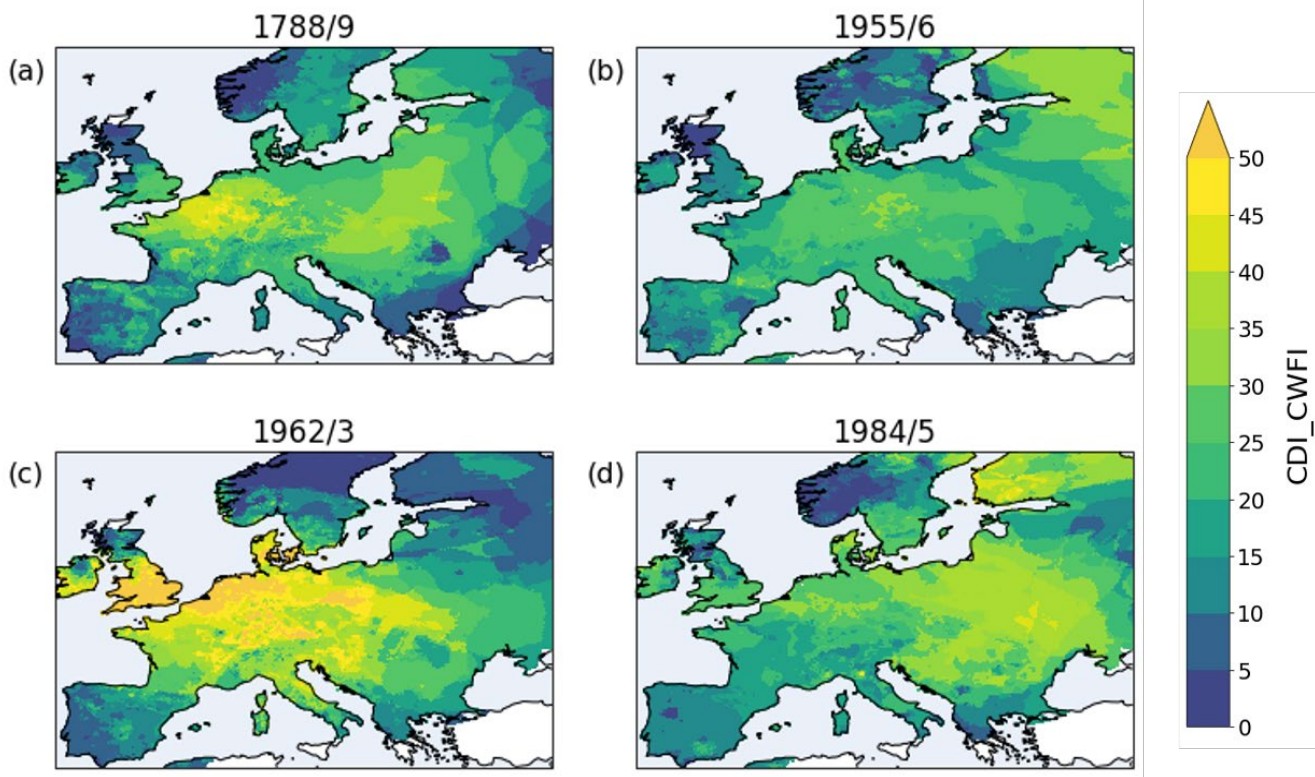

**Figure 8.** Cold-spell days index with reference to the 10th percentile of the daily mean temperatures for the period 1950-2020. The cdi_cwfi is calculated over the winter months NDJF for the following extreme winters: 1788/9 (a), 1955/6 (b), 1962/3 (c), and 1984/5 (d). The first of these is from the post-processed temperature reconstruction while information for the 20th-century winters is taken from the E-OBS dataset.