# Peer review of "Statistical reconstruction of daily temperature and sea-level pressure in Europe for the severe winter 1788/9"

_Climate of the Past, 2022_

## Author Response (AR1)

**Reviewer #1**

Thank you very much for your Review. Below are the replies to the comments raised.

Comment 1) The annual cycle is removed by filtering out the annual frequencies. However, the analogs themselves are chosen from candidates with a calendar date within a temporal window centered on the target. I wonder whether the filtering of the annual cycle is really necessary, since all analogs candidates are located in the same 'season' as the target. I do not think the filtering is damaging, but in my view it is not necessary. Perhaps the authors may like to add a couple of sentences to inform the reader

Reply 1) The reviewer notes that the subtraction of the annual cycle may not be necessary since anyway a seasonal window is specified. Note that this only concerns temperature (pressure is not deseasonalized). The reason for doing it is that this increases the size of the analog pool. This arguably does not matter for most cases, but it does for extremes, as for this extreme winter 1788/89.

In the revised manuscript we added (l. 205): "Note that deseasonalizing concerns only temperature and not pressure. The subtraction of the annual cycle increases the size of the analog pool, which might make it easier to reproduce extremes."
and specified the previous sentence: "For the spatial fields, the seasonality is removed …" -> "For the spatial temperature fields, the seasonality is removed…"

#################

Comment 2) Background state in the EnKF. The background state is chose as the best analog. I also wonder whether this is consistent with the calculation of the covariance matrix using all n-nearest neighbour analogs. It seems to me more logical to choose either the average of all n-analogs or possibly the member of the analog ensemble with median distance. Again, perhaps the auhors may want to comment on this

Reply 2) We prefer to work with one analog as this is physically consistent, while an average of analogs is not necessarily physically consistent. Of course, we could choose all n closest analogs as background and update them using the covariance matrix of n analogs and then do the ensemble mean (which again might not be physically consistent). We will discuss this in the revised manuscript.

In the revised manuscript we added (l. 242): "However, the best analogue is physical consistent while the average of the best n analogues is not."

#################

Comment 3) The Kalman filter set-up is generally used to combine two independent estimations, for instance one from a model run and one from a noisy observation. Both need to be independent for the method to be statistically sound. Here, however, both estimations are not independent: one is the best analog, which uses the observations, and the second is the observation itself. Thus, the separation is not clean, if I am not mistaken.

I would not be very picky here, since the authors test their results with independent observations and the method, pragmatically, indeed works: the EnSK is able to improve the

analog-based estimation. However, the more theory-inclined reader may frown upon this dependency. The authors may again want to include a warning or a comment.

Reply 3) The reviewer is correct that the background uses the observations; it is basically an interpolation of observations. However, the target day (and surrounding) is excluded from the pool of analogues, so we never assimilate observations from the same date in the evaluation experiments.

##################

Comment 4) 'The RMSE also shows an improvement from 3.4 to 2.7 °C, as does the mean bias from 0.67 to -0.13'

Reply 4) Thanks the unit is °C. We changed that.

**Reviewer #2**

General comment on re-writing and re-structuring of the paper: Having received no decision from the Editor, we have not restructured the paper.

Comment: "Please be careful of terminology when describing the method, I'm still not quite sure if 'post-processing' and EnKF mean the same thing, or whether either of them is included in ARM."

Reply:  We will better define our terminology. Our approach goes back to Pfister et al. (2019), where we applied EnKF to update temperature and quantile mapping to debias precipitation data and used "post-processing" as a summary term for both methods. We would like to keep the expression for consistency with that paper, but we now expand it to "post-processing by means of Ensemble Kalman Fitting"

lines 240, 369, 377: "post-processing" -> "post-processing by means of Ensemble Kalman Fitting"